# Towards Deeper Graph Neural Networks with Differentiable Group Normalization

**Kaixiong Zhou**
Texas A&M University
zkxiong@tamu.edu

**Xiao Huang**
The Hong Kong Polytechnic University
xiaohuang@comp.polyu.edu.hk

**Yuening Li**
Texas A&M University
liyuening@tamu.edu

**Daochen Zha**
Texas A&M University
daochen.zha@tamu.edu

**Rui Chen**
Samsung Research America
rui.chen1@samsung.com

**Xia Hu**
Texas A&M University
xiahu@tamu.edu

## Abstract

Graph neural networks (GNNs), which learn the representation of a node by aggregating its neighbors, have become an effective computational tool in downstream applications. Over-smoothing is one of the key issues which limit the performance of GNNs as the number of layers increases. It is because the stacked aggregators would make node representations converge to indistinguishable vectors. Several attempts have been made to tackle the issue by bringing linked node pairs close and unlinked pairs distinct. However, they often ignore the intrinsic community structures and would result in sub-optimal performance. The representations of nodes within the same community/class need be similar to facilitate the classification, while different classes are expected to be separated in embedding space. To bridge the gap, we introduce two over-smoothing metrics and a novel technique, i.e., differentiable group normalization (DGN). It normalizes nodes within the same group independently to increase their smoothness, and separates node distributions among different groups to significantly alleviate the over-smoothing issue. Experiments on real-world datasets demonstrate that DGN makes GNN models more robust to over-smoothing and achieves better performance with deeper GNNs.

## 1 Introduction

Graph neural networks (GNNs) [1, 2, 3] have emerged as a promising tool for analyzing networked data, such as biochemical networks [4, 5], social networks [6, 7], and academic networks [8, 9]. The successful outcomes have led to the development of many advanced GNNs, including graph convolutional networks [10], graph attention networks [11], and simple graph convolution networks [12].

Besides the exploration of graph neural network variants in different applications, understanding the mechanism and limitation of GNNs is also a crucial task. The core component of GNNs, i.e., a neighborhood aggregator updating the representation of a node iteratively via mixing itself with its neighbors' representations [6, 13], is essentially a low-pass smoothing operation [14]. It is in line with graph structures since the linked nodes tend to be similar [15]. It has been reported that, as the number of graph convolutional layers increases, all node representations over a graph will converge to indistinguishable vectors, and GNNs perform poorly in downstream applications [16, 17]. It is recognized as an over-smoothing issue. Such an issue prevents GNN models from going deeper to exploit the multi-hop neighborhood structures and learn better node representations.

A lot of efforts have been devoted to alleviating the over-smoothing issue, such as regularizing the node distance [18], node/edge dropping [19, 20], batch and pair normalizations [21, 22, 23].

Most of existing studies focused on measuring the over-smoothing based on node pair distances. By using these measurements, representations of linked nodes are forced to be close to each other, while unlinked pairs are separated. Unfortunately, the global graph structures and group/community characteristics are ignored, which leads to sub-optimal performance. For example, to perform node classification, an ideal solution is to assign similar vectors to nodes in the same class, instead of only the connected nodes. In the citation network Pubmed [24], $36\%$ of unconnected node pairs belong to the same class. These node pairs should instead have a small distance to facilitate node classification. Thus, we are motivated to tackle the over-smoothing issue in GNNs from a group perspective.

Given the complicated group structures and characteristics, it remains a challenging task to tackle the over-smoothing issue in GNNs. First, the formation of over-smoothing is complex and related to both local node relations and global graph structures, which makes it hard to measure and quantify. Second, the group information is often not directly available in real-world networks. This prevents existing tools such as group normalization being directly applied to solve our problem [25]. For example, while the group of adjacent channels with similar features could be directly accessed in convolutional neural networks [26], it is nontrivial to cluster a network in a suitable way. The node clustering needs to be in line with the embeddings and labels, during the dynamic learning process.

To bridge the gap, in this paper, we perform a quantitative study on the over-smoothing in GNNs from a group perspective. We aim to answer two research questions. First, how can we precisely measure the over-smoothing in GNNs? Second, how can we handle over-smoothing in GNNs? Through exploring these questions, we make three significant contributions as follows.

- Present two metrics to quantify the over-smoothing in GNNs: (1) Group distance ratio, clustering the network and measuring the ratio of inter-group representation distance over intra-group one; (2) Instance information gain, treating node instance independently and measuring the input information loss during the low-pass smoothing.

- Propose differentiable group normalization to significantly alleviate over-smoothing. It softly clusters nodes and normalizes each group independently, which prevents distinct groups from having close node representations to improve the over-smoothing metrics.

- Empirically show that deeper GNNs, when equipped with the proposed differentiable group normalization technique, yield better node classification accuracy.

## 2 Quantitative Analysis of Over-smoothing Issue

In this work, we use the semi-supervised node classification task as an example and illustrate how to handle the over-smoothing issue. A graph is represented by $G = \{\mathcal{V}, \mathcal{E}\}$, where $\mathcal{V}$ and $\mathcal{E}$ represent the sets of nodes and edges, respectively. Each node $v \in \mathcal{V}$ is associated with a feature vector $x_v \in \mathbb{R}^d$ and a class label $y_v$. Given a training set $\mathcal{V}_l$ accompanied with labels, the goal is to classify the nodes in the unlabeled set $\mathcal{V}_u = \mathcal{V} \setminus \mathcal{V}_l$ via learning the mapping function based on GNNs.

### 2.1 Preliminaries

Following the message passing strategy [27], GNNs update the representation of each node via aggregating itself and its neighbors' representations. Mathematically, at the $k$-th layer, we have,

$$N_v^{(k)} = \mathrm{AGG}(\{a_{vv'}^{(k)} W^{(k)} h_{v'}^{(k-1)} : v' \in \mathcal{N}(v)\}), \quad h_v^{(k)} = \mathrm{COM}(a_{vv}^{(k)} W^{(k)} h_v^{(k-1)}, N_v^{(k)}). \quad (1)$$

$N_v^{(k)}$ and $h_v^{(k)}$ denote the aggregated neighbor embedding and embedding of node $v$, respectively. We initialize $h_v^{(0)} = x_v$. $\mathcal{N}(v) = \{v'|e_{v,v'} \in \mathcal{E}\}$ represents the set of neighbors for node $v$, where $e_{v,v'}$ denotes the edge that connects nodes $v$ and $v'$. $W^{(k)}$ denotes the trainable matrix used to transform the embedding dimension. $a_{vv'}^{(k)}$ is the link weight over edge $e_{v,v'}$, which could be determined based on the graph topology or learned by an attention layer. Symbol AGG denotes the neighborhood aggregator usually implemented by a summation pooling. To update node $v$, function COM is applied to combine neighbor information and node embedding from the previous layer. It is observed that the weighted average in Eq. (1) smooths node embedding with its neighbors to make them similar. For a full GNN model with $K$ layers, the final node representation is given by $h_v = h_v^{(K)}$, which captures the neighborhood structure information within $K$ hops.

## 2.2 Measuring Over-smoothing with Group Structures

In GNNs, the neighborhood aggregation strategy smooths nodes' representations over a graph [14]. It will make the representations of nodes converge to similar vectors as the number of layers $K$ increases. This is called the over-smoothing issue, and would cause the performance of GNNs deteriorates as $K$ increases. To address the issue, the first step is to measure and quantify the over-smoothing [18, 20]. Measurements in existing work are mainly based on the distances between node pairs [19, 23]. A small distance means that a pair of nodes generally have undistinguished representation vectors, which might triggers the over-smoothing issue.

However, the over-smoothing is also highly related to global graph structures, which have not been taken into consideration. For some unlinked node pairs, we would need their representations to be close if they locate in the same class/community, to facilitate the node classification task. Without the specific group information, the metrics based on pair distances may fail to indicate the over-smoothing. Thus, we propose two novel over-smoothing metrics, i.e., group distance ratio and instance information gain. They quantify the over-smoothing from global (communities/classes/groups) and local (node individuals) views, respectively.

**Definition 1 (Group Distance Ratio).** Suppose that there are $C$ classes of node labels. We intuitively cluster nodes of the same class label into a group to formulate the labeled node community. Formally, let $\boldsymbol{L}_i = \{h_{iv}\}$ denote the group of representation vectors, where node $v$ is associated with label $i$. We have a series of labeled groups $\{\boldsymbol{L}_1, \cdots, \boldsymbol{L}_C\}$. Group distance ratio $R_{\text{Group}}$ measures the ratio of inter-group distance over intra-group distance in the Euclidean space. We have:

$$R_{\text{Group}} = \frac{\frac{1}{(C-1)^2} \sum_{i \neq j} (\frac{1}{|\boldsymbol{L}_i||\boldsymbol{L}_j|} \sum_{h_{iv} \in \boldsymbol{L}_i} \sum_{h_{jv'} \in \boldsymbol{L}_j} ||h_{iv} - h_{jv'}||_2)}{\frac{1}{C} \sum_i (\frac{1}{|\boldsymbol{L}_i|^2} \sum_{h_{iv}, h_{iv'} \in \boldsymbol{L}_i} ||h_{iv} - h_{iv'}||_2)}, \qquad (2)$$

where $|| \cdot ||_2$ denotes the L2 norm of a vector and $| \cdot |$ denotes the set cardinality. The numerator (denominator) represents the average of pairwise representation distances between two different groups (within a group). One would prefer to reduce the intra-group distance to make representations of the same class similar, and increase the inter-group distance to relieve the over-smoothing issue. On the contrary, a small $R_{\text{Group}}$ leads to the over-smoothing issue where all groups are mixed together, and the intra-group distance is maintained to hinder node classification.

**Definition 2 (Instance Information Gain).** In an attributed network, a node's feature decides its class label to some extent. We treat each node instance independently, and define instance information gain $G_{\text{Ins}}$ as how much input feature information is contained in the final representation. Let $\mathcal{X}$ and $\mathcal{H}$ denote the random variables of input feature and representation vector, respectively. We define their probability distributions with $P_{\mathcal{X}}$ and $P_{\mathcal{H}}$, and use $P_{\mathcal{X}\mathcal{H}}$ to denote their joint distribution. $G_{\text{Ins}}$ measures the dependency between node feature and representation via their mutual information:

$$G_{\text{Ins}} = I(\mathcal{X}; \mathcal{H}) = \sum_{x_v \in \mathcal{X}, h_v \in \mathcal{H}} P_{\mathcal{X}\mathcal{H}}(x_v, h_v) \log \frac{P_{\mathcal{X}\mathcal{H}}(x_v, h_v)}{P_{\mathcal{X}}(x_v) P_{\mathcal{H}}(h_v)}. \qquad (3)$$

We list the details of variable definitions and mutual information calculation in the context of GNNs in Appendix. With the intensification of the over-smoothing issue, nodes average the neighborhood information and lose their self features, which leads to a small value of $G_{\text{Ins}}$.

## 2.3 Illustration of Proposed Over-smoothing Metrics

Based on the two proposed metrics, we take simple graph convolution networks (SGC) as an example, and analyze the over-smoothing issue on Cora dataset [24]. SGC simplifies the model through removing all the trainable weights between layers to avoid the potential of overfitting [12]. So the over-smoothing issue would be the major cause of performance dropping in SGC. As shown by the red lines in Figure 1, the graph convolutions first exploit neighborhood information to improve test accuracy up to $K = 5$, after which the over-smoothing issue starts to worsen the performance. At the same time, instance information gain $G_{\text{Ins}}$ and group distance ratio $R_{\text{Group}}$ decrease due to the over-smoothing issue. For the extreme case of $K = 120$, the input features are filtered out and all groups of nodes converge to the same representation vector, leading to $G_{\text{Ins}} = 0$ and $R_{\text{Group}} = 1$, respectively. Our metrics quantify the smoothness of node representations based on group structures, but also have the similar variation tendency with test accuracy to indicate it well.

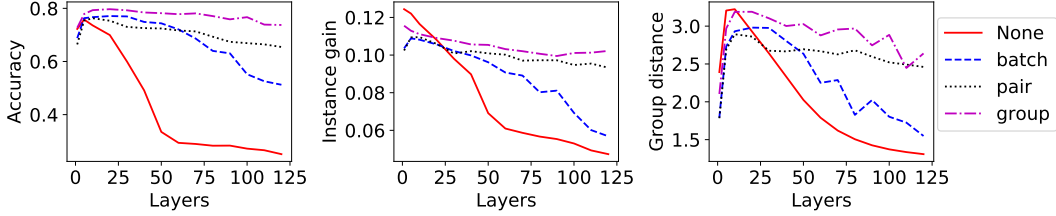

Figure 1: The test accuracy, instance information gain, and group distance ratio of SGC on Cora. We compare differentiable group normalization with none, batch and pair normalizations.

## 3 Differentiable Group Normalization

We start with a graph-regularized optimization problem [10, 18]. To optimize the over-smoothing metrics of $G_{\text{Ins}}$ and $R_{\text{Group}}$, one traditional approach is to minimize the loss function:

$$\mathcal{L} = \mathcal{L}_0 - G_{\text{Ins}} - \lambda R_{\text{Group}}. \tag{4}$$

$\mathcal{L}_0$ denotes the supervised cross-entropy loss w.r.t. representation probability vectors $h_v \in \mathbb{R}^{C \times 1}$ and class labels. $\lambda$ is a balancing factor. The goal of optimization problem Eq. (4) is to learn node representations close to the input features and informative for their class labels. Considering the labeled graph communities, it also improves the intra-group similarity and inter-group distance. However, it is non-trivial to optimize this objective function due to the non-derivative of non-parametric statistic $G_{\text{Ins}}$ [28, 29] and the expensive computation of $R_{\text{Group}}$.

### 3.1 Proposed Technique for Addressing Over-smoothing

Instead of directly optimizing regularized problem in Eq. (4), we propose the differentiable group normalization (DGN) applied between graph convolutional layers to normalize the node embeddings group by group. The key intuition is to cluster nodes into multiple groups and then normalize them independently. Consider the labeled node groups (or communities) in networked data. The node embeddings within each group are expected to be rescaled with a specific mean and variance to make them similar. Meanwhile, the embedding distributions from different groups are separated by adjusting their means and variances. We develop an analogue with the group normalization in convolutional neural networks (CNNs) [25], which clusters a set of adjacent channels with similar characteristics into a group and treats it independently. Compared with standard CNNs, the challenge in designing DGN is how to cluster nodes in a suitable way. The clustering needs to be in line with the embedding and labels, during the dynamic learning process.

We address this challenge by learning a cluster assignment matrix, which softly maps nodes with close embeddings into a group. Under the supervision of training labels, the nodes close in the embedding space tend to share a common label. To be specific, we first describe how DGN clusters and normalizes nodes in a group-wise fashion given an assignment matrix. After that, we discuss how to learn the assignment matrix to support differentiable node clustering.

**Group Normalization.** Let $H^{(k)} = [h_1^{(k)}, \cdots, h_n^{(k)}]^T \in \mathbb{R}^{n \times d^{(k)}}$ denote the embedding matrix generated from the $k$-th graph convolutional layer. Taking $H^{(k)}$ as input, DGN softly assigns nodes into groups and normalizes them independently to output a new embedding matrix for the next layer. Formally, we define the number of groups as $G$, and denote the cluster assignment matrix by $S^{(k)} \in \mathbb{R}^{n \times G}$. $G$ is a hyperparameter that could be tuned per dataset. The $i$-th column of $S^{(k)}$, i.e., $S^{(k)}[:, i]$, indicates the assignment probabilities of nodes in a graph to the $i$-th group. Supposing that $S^{(k)}$ has already been computed, we cluster and normalize nodes in each group as follows:

$$H_i^{(k)} = S^{(k)}[:, i] \circ H^{(k)} \in \mathbb{R}^{n \times d^{(k)}}; \quad \tilde{H}_i^{(k)} = \gamma_i \left( \frac{H_i^{(k)} - \mu_i}{\sigma_i} \right) + \beta_i \in \mathbb{R}^{n \times d^{(k)}}. \tag{5}$$

Symbol $\circ$ denotes the row-wise multiplication. The left part in the above equation represents the soft node clustering for group $i$, whose embedding matrix is given by $H_i^{(k)}$. The right part performs the standard normalization operation. In particular, $\mu_i$ and $\sigma_i$ denote the vectors of running mean

and standard deviation of group $i$, respectively, and $\gamma_i$ and $\beta_i$ denote the trainable scale and shift vectors, respectively. Given the input embedding $H^{(k)}$ and the series of normalized embeddings $\{\tilde{H}_1^{(k)}, \cdots, \tilde{H}_G^{(k)}\}$, DGN generates the final embedding matrix $\tilde{H}^{(k)}$ for the next layer as follows:

$$\tilde{H}^{(k)} = H^{(k)} + \lambda \sum_{i=1}^{G} \tilde{H}_i^{(k)} \in \mathbb{R}^{n \times d^{(k)}}. \tag{6}$$

$\lambda$ is a balancing factor as mentioned before. Inspecting the loss function in Eq. (4), DGN utilizes components $H^{(k)}$ and $\sum_{i=1}^{G} \tilde{H}_i^{(k)}$ to improve terms $G_{\text{Ins}}$ and $R_{\text{Group}}$, respectively. In particular, we preserve the input embedding $H^{(k)}$ to avoid over-normalization and keep the input feature of each node to some extent. Note that the linear combination of $H^{(k)}$ in DGN is different from the skip connection in GNN models [30, 31], which instead connects the embedding output $H^{(k-1)}$ from the last layer. The technique of skip connection could be included to further boost the model performance. Group normalization $\sum_{i=1}^{G} \tilde{H}_i^{(k)}$ rescales the node embeddings within each group independently to make them similar. Ideally, we assign the close node embeddings with a common label to a group. Node embeddings of the group are then distributed closely around the corresponding running mean. Thus for different groups associate with distinct node labels, we disentangle their running means and separates the node embedding distributions. By applying DGN between the successive graph convolutional layers, we are able to optimize Problem (4) to mitigate the over-smoothing issue.

**Differentiable Clustering.** We apply a linear model to compute the cluster assignment matrix $S^{(k)}$ used in Eq. (5). The mathematical expression is given by:

$$S^{(k)} = \text{softmax}(H^{(k)} U^{(k)}). \tag{7}$$

$U^{(k)} \in \mathbb{R}^{d^{(k)} \times G}$ denotes the trainable weights for a DGN module applied after the $k$-th graph convolutional layer. $\text{softmax}$ function is applied in a row-wise way to produce the normalized probability vector w.r.t all the $G$ groups for each node. Through the inner product between $H^{(k)}$ and $U^{(k)}$, the nodes with close embeddings are assigned to the same group with a high probability. Here we give a simple and effective way to compute $S^{(k)}$. Advanced neural networks could be applied.

**Time Complexity Analysis.** Suppose that the time complexity of embedding normalization at each group is $\mathcal{O}(T)$, where $T$ is a constant depending on embedding dimension $d^{(k)}$ and node number $n$. The time cost of group normalization $\sum_{i=1}^{G} \tilde{H}_i^{(k)}$ is $\mathcal{O}(GT)$. Both the differentiable clustering (in Eq. (5)) and the linear model (in Eq. (7)) have a time cost of $\mathcal{O}(nd^{(k)}G)$. Thus the total time complexity of a DGN layer is given by $\mathcal{O}(nd^{(k)}G + GT)$, which linearly increases with $G$.

**Comparison with Prior Work.** To the best of our knowledge, the existing work mainly focuses on analyzing and improving the node pair distance to relieve the over-smoothing issue [18, 20, 23]. One of the general solutions is to train GNN models regularized by the pair distance [18]. Recently, there are two related studies applying batch normalization [21] or pair normalization [23] to keep the overall pair distance in a graph. The pair normalization is a "slim" realization of the batch normalization by removing the trainable scale and shift. However, the metric of pair distance and the resulting techniques ignore the global graph structure, and may achieve sub-optimal performance in practice. In this work, we measure the over-smoothing based on communities/groups and independent node instances. We then formulate the problem in Eq. (4) to optimize the proposed metrics, and propose DGN to solve it in an efficient way, which in turn addresses the over-smoothing issue. Parallel to this research, the similar group normalization approaches of attentive normalization (AN) [32] and attentive context normalization (ACN) [33] are used for the semantic segmentation at computer vision domain. Specially, ACN has only one group and AN additionally samples random groups during model inference. They are not in line with the transductive node classification task where the underlying graph has a series of fixed community structures. Furthermore, the motivations for our DGN and AN/ACN are different. While AN/ACN target at capturing the long-range relations between pixels, DGN intends to improve the distance between different groups to mitigate the over-smoothing.

### 3.2 Evaluating Differentiable Group Normalization on Attributed Graphs

We apply DGN to the SGC model to validate its effectiveness in relieving the over-smoothing issue. Furthermore, we compare with the other two available normalization techniques used upon GNNs,

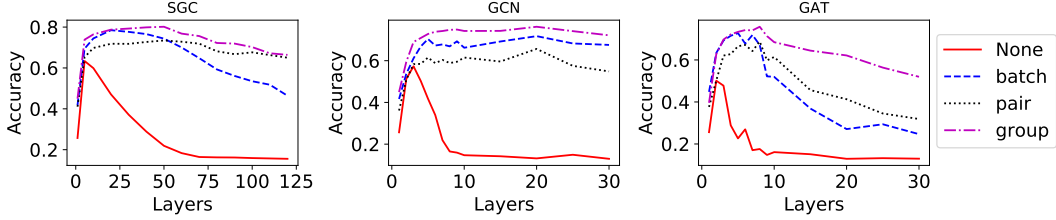

Figure 2: The test accuracies of SGC, GCN, and GAT models on Cora with missing features. We compare differentiable group normalization with none, batch and pair normalizations.

i.e., batch normalization and pair normalization. As shown in Figure 1, the test accuracy of DGN remains stable with the increase in the number of layers. By preserving the input embedding and normalizing node groups independently, DGN achieves superior performance in terms of instance information gain as well as group distance ratio. The promising results indicate that our DGN tackles the over-smoothing issue more effectively, compared with none, batch and pair normalizations.

It should be noted that, the highest accuracy of $79.7\%$ is achieved with DGN when $K = 20$. This observation contradicts with the common belief that GNN models work best with a few layers on current benchmark datasets [34]. With the integration of advanced techniques, such as DGN, we are able to exploit deeper GNN architectures to unleash the power of deep learning in network analysis.

### 3.3 Evaluation in Scenario with Missing Features

To further illustrate that DGN could enable us to achieve better performance with deeper GNN architectures, we apply it to a more complex scenario. We assume that the attributes of nodes in the test set are missing. It is a common scenario in practice [23]. For example, in social networks, new users are often lack of profiles and tags [35]. To perform prediction tasks on new users, we would rely on the node attributes of existing users and their connections to new users. In such a scenario, we would like to apply more layers to exploit the neighborhood structure many hops away to improve node representation learning. Since the over-smoothing issue gets worse with the increasing of layer numbers, the benefit of applying normalization will be more obvious in this scenario.

We remove the input features of both validation and test sets in Cora, and replace them with zeros [23]. Figure 2 presents the results on three widely-used models, i.e., SGC, graph convolutional networks (GCN), and graph attention networks (GAT). Due to the over-smoothing issue, GNN models without any normalization fail to distinguish nodes quickly with the increasing number of layers. In contrast, the normalization techniques reach their highest performance at larger layer numbers, after which they drop slowly. We observe that DGN obtains the best performance with $50$, $20$, and $8$ layers for SGC, GCN, and GAT, respectively. These layer numbers are significantly larger than those of the widely-used shallow models (e.g., two or three layers).

## 4 Experiments

We now empirically evaluate the effectiveness and robustness of DGN on real-world datasets. We aim to answer three questions as follows. **Q1:** Compared with the state-of-the-art normalization methods, can DGN alleviate the over-smoothing issue in GNNs in a better way? **Q2:** Can DGN help GNN models achieve better performance by enabling deeper GNNs? **Q3:** How do the hyperparameters influence the performance of DGN? The implementation of our approaches is publicly available at https://github.com/Kaixiong-Zhou/DGN.

### 4.1 Experiment Setup

**Datasets.** Joining the practice of previous work, we evaluate GNN models by performing the node classification task on four datasets: Cora, Citeseer, Pubmed [24], and CoauthorCS [36]. We also create graphs by removing features in validation and test sets. The dataset statistics are in Appendix.

**Implementations.** Following the previous settings, we choose the hyperparameters of GNN models and optimizer as follows. We set the number of hidden units to 16 for GCN and GAT models. The

| Dataset | Model | Layers 2/5 | | | | Layers 15/60 | | | | Layers 30/120 | | | | #K |
|---|---|---|---|---|---|---|---|---|---|---|---|---|---|---|
| | | NN | BN | PN | DGN | NN | BN | PN | DGN | NN | BN | PN | DGN | |
| Cora | GCN | **82.2** | 73.9 | 71.0 | 82.0 | 18.1 | 70.3 | 67.2 | **75.2** | 13.1 | 67.2 | 64.3 | **73.2** | 2 |
| | GAT | 80.9 | 77.8 | 74.4 | **81.1** | 16.8 | 33.1 | 49.6 | **71.8** | 13.0 | 25.0 | 30.2 | **51.3** | 2 |
| | SGC | 75.8 | 76.3 | 75.4 | **77.9** | 29.4 | 72.1 | 71.7 | **77.8** | 25.1 | 51.2 | 65.5 | **73.7** | 20 |
| Citeseer | GCN | **70.6** | 51.3 | 60.5 | 69.5 | 15.2 | 46.9 | 46.7 | **53.1** | 9.4 | 47.9 | 47.1 | **52.6** | 2 |
| | GAT | **70.2** | 61.5 | 62.0 | 69.3 | 22.6 | 28.0 | 41.4 | **52.6** | 7.7 | 21.4 | 33.3 | **45.6** | 2 |
| | SGC | **69.6** | 58.8 | 64.8 | 69.5 | **66.3** | 50.5 | 65.0 | 63.4 | 60.8 | 47.3 | 63.1 | **64.7** | 30 |
| Pubmed | GCN | 79.3 | 74.9 | 71.1 | **79.5** | 22.5 | 73.7 | 70.6 | **76.1** | 18.0 | 70.4 | 70.4 | **76.9** | 2 |
| | GAT | **77.8** | 76.2 | 72.4 | 77.5 | 37.5 | 56.2 | 68.8 | **75.9** | 18.0 | 46.6 | 58.2 | **73.3** | 5 |
| | SGC | 71.5 | 76.5 | 75.8 | **76.8** | 34.2 | 75.2 | 77.1 | **77.4** | 23.1 | 71.6 | 76.7 | **77.1** | 10 |
| Coauthors | GCN | 92.3 | 86.0 | 77.8 | **92.3** | 72.2 | 78.5 | 69.5 | **83.7** | 3.3 | **84.7** | 64.5 | 84.4 | 1 |
| | GAT | 91.5 | 89.4 | 85.9 | **91.8** | 6.0 | 77.7 | 53.1 | **84.5** | 3.3 | 16.7 | 48.1 | **75.5** | 1 |
| | SGC | 89.9 | 88.7 | 86.0 | **90.2** | 10.2 | 59.7 | 76.4 | **81.3** | 5.8 | 30.5 | 52.6 | **60.8** | 1 |

Table 1: Test accuracy in percentage on the attributed networks. Layers $a/b$ denote the layer number $a$ in models GCN & GAT and that of $b$ in model SGC. #$K$ denotes the optimal layer numbers where DGN achieves the highest performance.

number of attention heads in GAT is 1. Since a larger parameter size in GCN and GAT may lead to overfitting and affects the study of over-smoothing issue, we compare normalization methods by varying the number of layers $K$ in $\{1, 2, \cdots, 10, 15, \cdots, 30\}$. For SGC, we increase the testing range and vary $K$ in $\{1, 5, 10, 20, \cdots, 120\}$. We train with a maximum of 1000 epochs using the Adam optimizer [37] and early stopping. Weights in GNN models are initialized with Glorot algorithm [38]. We use the following sets of hyperparameters for Citeseer, Cora, CoauthorCS: 0.6 (dropout rate), $5 \cdot 10^{-4}$ (L2 regularization), $5 \cdot 10^{-3}$ (learning rate), and for Pubmed: 0.6 (dropout rate), $1 \cdot 10^{-3}$ (L2 regularization), $1 \cdot 10^{-2}$ (learning rate). We run each experiment 5 times and report the average.

**Baselines.** We compare with none normalization (NN), batch normalization (BN) [21, 22] and pair normalization (PN) [23]. Their technical details are listed in Appendix.

**DGN Configurations.** The key hyperparameters include group number $G$ and balancing factor $\lambda$. Depending on the number of class labels, we apply 5 groups to Pubmed and 10 groups to the others. The criterion is to use more groups to separate representation distributions in networked data accompanied with more class labels. $\lambda$ is tuned on validation sets to find a good trade-off between preserving input features and group normalization. We introduce the selection of $\lambda$ in Appendix.

## 4.2 Experiment Results

**Studies on alleviating the over-smoothing problem.** To answer **Q1**, Table 1 summarizes the results of applying the different normalization techniques to GNN models on all the datasets. We report the performances of GCN and GAT with 2/15/30 layers, and SGC with 5/60/120 layers due to space limit. We provide the test accuracies, instance information gain and group distance ratio under all depths in Appendix. Given the same layers, it can be observed that DGN almost outperforms the other normalizations for all cases. DGN significantly slows down the performance dropping with the increment of layers, and alleviates the over-smoothing issue. That is because the self-preserved component $H^{(k)}$ in Eq. (6) keeps the informative input features and avoids over-normalization to distinguish the different nodes. This component is especially crucial for models with a few layers since the over-smoothing issue has not appeared. The other group normalization component in Eq. (6) processes each group of nodes independently. It disentangles the representation similarity between groups, and hence reduces the over-smoothness of nodes over a graph accompanied with graph convolutions. The optimal layer numbers #$K$ of SGC are generally larger than those of other GNN models, since the redundant weights are removed to avoid the over-fitting issue.

**Studies on enabling deeper and better GNNs.** To answer **Q2**, we compare all of the concerned normalization methods over GCN, GAT, and SGC in the scenario with missing features. As we have discussed, the normalization techniques will show their power in relieving the over-smoothing issue and exploring the deeper architectures especially for this scenario. In Table 2, Acc represents the best test accuracy yielded by model equipped with the optimal layer number #$K$. It is shown that

| Model | Norm | Cora Acc | Cora #K | Citeseer Acc | Citeseer #K | Pubmed Acc | Pubmed #K | CoauthorCS Acc | CoauthorCS #K | Improvement% |
|-------|------|----------|---------|--------------|-------------|------------|-----------|----------------|---------------|--------------|
| GCN | NN | 57.3 | 3 | 44.0 | 6 | 36.4 | 4 | 67.3 | 3 | 42.2 |
|  | BN | 71.8 | 20 | 45.1 | 25 | 70.4 | 30 | 82.7 | 30 | 5.2 |
|  | PN | 65.6 | 20 | 43.6 | 25 | 63.1 | 30 | 63.5 | 4 | 19.2 |
|  | DGN | **76.3** | **20** | **50.2** | **30** | **72.0** | **30** | **83.7** | **25** | - |
| GAT | NN | 50.1 | 2 | 40.8 | 4 | 38.5 | 4 | 63.7 | 3 | 51.0 |
|  | BN | 72.7 | 5 | 48.7 | 5 | 60.7 | 4 | 80.5 | 6 | 9.8 |
|  | PN | 68.8 | 8 | 50.3 | 6 | 63.2 | 20 | 66.6 | 3 | 14.7 |
|  | DGN | **75.8** | **8** | **54.5** | **5** | **72.3** | **20** | **83.6** | **15** | - |
| SGC | NN | 63.4 | 5 | 51.2 | 40 | 63.7 | 5 | 71.0 | 5 | 20.1 |
|  | BN | 78.5 | 20 | 50.4 | 20 | 72.3 | 50 | 84.4 | 20 | 6.2 |
|  | PN | 73.4 | 50 | 58.0 | 120 | 75.2 | 30 | 80.1 | 10 | 4.5 |
|  | DGN | **80.2** | **50** | **58.2** | **90** | **76.2** | **90** | **85.8** | **20** | - |

Table 2: The highest accuracy (%) and the accompanied optimal layers in the scenario with missing features. We calculate the average improvement achieved by DGN over each GNN framework.

| Dataset | Model | Layers 2 NN | Layers 2 BN | Layers 2 PN | Layers 2 DGN | Layers 16 NN | Layers 16 BN | Layers 16 PN | Layers 16 DGN | Layers 64 NN | Layers 64 BN | Layers 64 PN | Layers 64 DGN |
|---------|-------|-------------|-------------|-------------|--------------|--------------|--------------|--------------|---------------|--------------|--------------|--------------|---------------|
| Cora | GCNII* | 81.9 | 76.4 | 67.4 | **82.7** | 84.4 | 70.2 | 64.9 | **84.8** | 85.3 | 69.2 | 68.6 | **85.6** |

Table 3: Test accuracy in percentage on the attributed dataset Cora over GCNII backbone.

the values of $\#K$ are much larger than those in Table 1, which empirically validate the assumption that a deeper GNN architecture is particularly required in the missing-feature scenario to exploit the neighborhood information many hops away. More importantly, DGN significantly outperforms the other normalization methods on all the cases. The average improvements over NN, BN and PN achieved by DGN are 37.8%, 7.1% and 12.8%, respectively. Via clustering and disentangling the different groups, DGN tackles the over-smoothing issue to enable the exploration of more powerful deeper architectures. We present the comprehensive analyses in terms of test accuracy, instance information gain and group distance ratio under all depths in Appendix.

**Evaluation on deep GNN architecture.** As a general normalization layer, DGN could also be applied to the existing deep GNN architectures to further improve the node classification performance. Recently, model GCNII* [39] has been proposed to prevent the over-smoothing by the residual connection and the identity mapping, which achieves the new state-of-the-art results with a 64-layer architecture. To support answering **Q2**, we compare the different normalization approaches over the GCNII* backbone based on the provided implementation. Table 3 shows the test accuracies obtained on dataset Cora, where DGN consistently outperforms the other normalization approaches under the different network depths. Note that NN (i.e., none normalization is included) represents the original GCNII* model, which could be further improved with our DGN. These results validate that DGN is a general normalization module to relieve the over-smoothing issue for the different GNN architectures.

**Hyperparameter studies.** We study the impact of hyperparameters, group number $G$ and balancing factor $\lambda$, on DGN in order to answer research question **Q3**. Over the GCN framework associated with 20 convolutional layers, we evaluate DGN by considering $G$ and $\lambda$ from sets $[1, 5, 10, 15, 20, 30]$ and $[0.001, 0.005, 0.01, 0.03, 0.05, 0.1]$, respectively. The left part in Figure 3 presents the test accuracy for each hyperparameter combination. We observe that: (i) The model performance is damaged greatly when $\lambda$ is close to zero (e.g., $\lambda = 0.001$). In this case, group normalization contributes slightly in DGN, resulting in over-smoothing in the GCN model. (ii) Model performance is not sensitive to the value of $G$, and an appropriate $\lambda$ value could be tuned to optimize the trade-off between instance gain and group normalization. It is because DGN learns to use the appropriate number of groups by end-to-end training. In particular, some groups might not be used as shown in the right part of Figure 3, at which only 6 out of 10 groups (denoted by black triangles) are adopted. (iii) Even when $G = 1$, DGN still outperforms BN by utilizing the self-preserved component to achieve an accuracy of 74.7%, where $\lambda = 0.1$. Via increasing the group number, the model performance could be further improved, e.g., the accuracy of 76.3% where $G = 10$ and $\lambda = 0.01$.

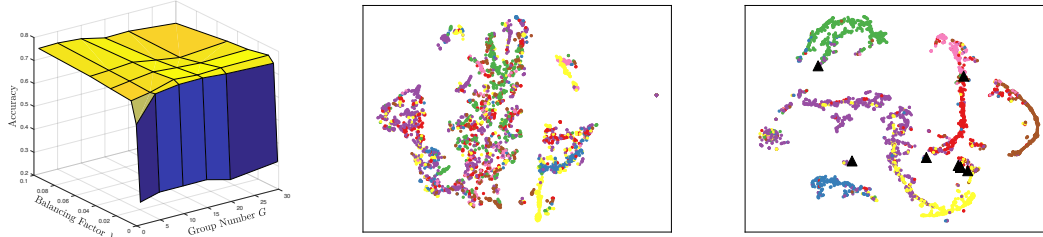

Figure 3: **Left**: Test accuracies of GCN with 20 layers on Cora with missing features, where hyperparameters $G$ and $\lambda$ are studied. **Middle**: Node representation visualization for GCN without normalization and with $K = 20$. **Right**: Node representation visualization for GCN with DGN layer and $K = 20$ (node colors represent classes, and black triangles denote the running means of groups).

**Node representation visualization.** We investigate how DGN clusters nodes into different groups to tackle the over-smoothing issue. The middle and right parts of Figure 3 visualize the node representations achieved by GCN models without normalization tool and with the DGN approach, respectively. It is observed that the node representations of different classes mix together when the layer number reaches 20 in the GCN model without normalization. In contrast, our DGN method softly assigns nodes into a series of groups, whose running means at the corresponding normalization modules are highlighted with black triangles. Through normalizing each group independently, the running means are separated to improve inter-group distances and disentangle node representations. In particular, we notice that the running means locate at the borders among different classes (e.g., the upper-right triangle at the border between red and pink classes). That is because the soft assignment may cluster nodes of two or three classes into the same group. Compared with batch or pair normalization, the independent normalization for each group only includes a few classes in DGN. In this way, we relieve the representation noise from other node classes during normalization, and improve the group distance ratio as illustrated in Appendix.

## 5 Conclusion

In this paper, we propose two over-smoothing metrics based on graph structures, i.e., group distance ratio and instance information gain. By inspecting GNN models through the lens of these two metrics, we present a novel normalization layer, DGN, to boost model performance against over-smoothing. It normalizes each group of similar nodes independently to separate node representations of different classes. Experiments on real-world classification tasks show that DGN greatly slowed down performance degradation by alleviating the over-smoothing issue. DGN enables us to explore deeper GNNs and achieve higher performance in analyzing attributed networks and the scenario with missing features. Our research will facilitate deep learning models for potential graph applications.

## Acknowledgements

This work is, in part, supported by NSF (#IIS-1750074, #IIS-1718840, and #IIS-1900990). The views, opinions, and/or findings contained in this paper are those of the authors and should not be interpreted as representing any funding agencies.

## Broader Impact

The successful outcome of this work will lead to advances in building up deep graph neural networks and dealing with complex graph-structured data. The developed metrics and algorithms have an immediate and strong impact on a number of fields, including (1) *Over-smoothing Quantitative Analysis*: GNN models tend to result in the over-smoothing issue with the increase in the number of layers. During the practical development of deeper GNN models, the proposed instance information gain and group distance ratio effectively indicate the over-smoothing issue, in order to push the model exploration toward a good direction. (2) *Deep GNN Modeling*: The proposed differentiable group normalization tool successfully tackles the over-smoothing issue and enables the modeling of

deeper GNN variants. It encourages us to fully unleash the power of deep learning in processing the networked data. (3) *Real-world Network Analytics Applications*: The proposed research will broadly shed light on utilizing deep GNN models in various applications, such as social network analysis, brain network analysis, and e-commerce network analysis. For such complex graph-structured data, deep GNN models can exploit the multi-hop neighborhood information to boost the task performance.

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
