[Supplementary Material · GroupNorm_Appendix_NeuIPS_2020.pdf]

# Appendix

**Kaixiong Zhou**
Texas A&M University
zkxiong@tamu.edu

**Xiao Huang**
The Hong Kong Polytechnic University
xiaohuang@comp.polyu.edu.hk

**Yuening Li**
Texas A&M University
liyuening@tamu.edu

**Daochen Zha**
Texas A&M University
daochen.zha@tamu.edu

**Rui Chen**
Samsung Research America
rui.chen1@samsung.com

**Xia Hu**
Texas A&M University
xiahu@tamu.edu

## 1   Dataset Statistics

For fair comparison with previous work, we perform the node classification task on four benchmark datasets, including Cora, Citeseer, Pubmed [1], and CoauthorCS [2]. They have been widely adopted to study the over-smoothing issue in GNNs [3, 4, 5, 6, 7]. The detailed statistics are listed in Table 1. To further illustrate that the normalization techniques could enable deeper GNNs to achieve better performance, we apply them to a more complex scenario with missing features. For these four benchmark datasets, we create the corresponding scenarios by removing node features in both validation and testing sets.

|  | Cora | Citeseer | Pubmed | CoauthorCS |
|---|---|---|---|---|
| #Nodes | 2708 | 3327 | 19717 | 18333 |
| #Edges | 5429 | 4732 | 44338 | 81894 |
| #Features | 1433 | 3703 | 500 | 6805 |
| #Classes | 7 | 6 | 3 | 15 |
| #Training Nodes | 140 | 120 | 60 | 600 |
| #Validation Nodes | 500 | 500 | 500 | 2250 |
| #Testing Nodes | 1000 | 1000 | 1000 | 15483 |

Table 1: Dataset statistics on Cora, Citeseer, Pubmed, and CoauthorCS.

## 2   Running Environment

All the GNN models and normalization approaches are implemented in PyTorch, and tested on a machine with 24 Intel(R) Xeon(R) CPU E5-2650 v4 @ 2.20GB processors, GeForce GTX-1080 Ti 12 GB GPU, and 128GB memory size. We implement the group normalization in a parallel way. Thus the practical time cost of our DGN is comparable to that of traditional batch normalization.

## 3   GNN Models

We test over three general GNN models to illustrate the over-smoothing issue, including graph convolutional networks (GCN) [8], graph attention networks (GAT) [9] and simple graph convolution (SGC) networks [10]. We list their neighbor aggregation functions in Table 2.

Considering the message passing strategy as shown by Eq. (1) in the main manuscript, we explain the key properties of GCN, GAT and SGC as follows. GCN merges the information from node itself and its neighbors weighted by vertices' degrees, where $a_{vv'}^{(k)} = 1./\sqrt{(|\mathcal{N}(v)| + 1) \cdot (|\mathcal{N}(v')| + 1)}$.

| Model | Neighbor aggregation function |
|---|---|
| GCN | $\mathrm{h}_v^{(k)} = \mathrm{ReLU}(\sum_{v'\in\mathcal{N}(v)\cup\{v\}} \frac{1}{\sqrt{(|\mathcal{N}(v)|+1)\cdot(|\mathcal{N}(v')|+1)}} W^{(k)} h_{v'}^{(k-1)})$ |
| GAT | $\mathrm{h}_v^{(k)} = \mathrm{ReLU}(\sum_{v'\in\mathcal{N}(v)\cup\{v\}} a_{vv'}^{(k)} W^{(k)} h_{v'}^{(k-1)})$ |
| SGC | $\mathrm{h}_v^{(k)} = \sum_{v'\in\mathcal{N}(v)\cup\{v\}} \frac{1}{\sqrt{(|\mathcal{N}(v)|+1)\cdot(|\mathcal{N}(v')|+1)}} h_{v'}^{(k-1)}$ |

Table 2: Neighbor aggregation function at a graph convolutional layer for GCN, GAT and SGC.

Functions AGG and COM are realized by a summation pooling. The activation function of ReLU is then applied to non-linearly transform the latent embedding. Based on GCN, GAT uses an additional attention layer to learn link weight $a_{vv'}^{(k)}$. GAT aggregates neighbors with the trainable link weights, and achieves significant improvements in a variety of applications. SGC is simplified from GCN by removing all trainable parameters $W^{(k)}$ and nonlinear activations between successive layers. It has been empirically shown that these simplifications do not negatively impact classification accuracy, and even relive the problems of over-fitting and vanishing gradients in deeper models.

## 4 Normalization Baselines

Batch normalization is first applied between the successive convolutional layers in CNNs [11]. It is extended to graph neural networks to improve node representation learning and generalization [12]. Taking embedding matrix $H^{(k)}$ as input after each layer, batch normalization scales the node representations using running mean and variance, and generates a new embedding matrix for the next graph convolutional layer. Formally, we have:

$$\tilde{H}^{(k)} = \gamma(\frac{H^{(k)} - \mu}{\sigma}) + \beta \in \mathbb{R}^{n\times d^{(k)}}.$$

$\mu$ and $\sigma$ denote the vectors of running mean and standard deviation, respectively; $\gamma$ and $\beta$ denote the trainable scale and shift vectors, respectively. Recently, pair normalization has been proposed to tackle the over-smoothing issue in GNNs, targeting at maintaining the average node pair distance over a graph [5]. Pair normalization is a simplifying realization of batch normalization by removing the trainable $\gamma$ and $\beta$. In this work, we augment each graph convolutional layer via appending a normalization module, in order to validate the effectiveness of normalization technique in relieving over-smoothing and enabling deeper GNNs.

## 5 Hyperparameter Tuning in DGN

The balancing factor, $\lambda$, is crucial to determine the trade-off between input feature preservation and group normalization in DGN. It needs to be tuned carefully as GNN models increase the number of layers. To be specific, we consider the candidate set $\{5\cdot10^{-4}, 1\cdot10^{-3}, 2\cdot10^{-3}, 3\cdot10^{-3}, 5\cdot10^{-3}, 1\cdot10^{-2}, 2\cdot10^{-2}, 3\cdot10^{-2}, 5\cdot10^{-2}\}$. For each specific model, we use a few epochs to choose the optimal $\lambda$ on the validation set, and then evaluate it on the testing set. We observe that the value of $\lambda$ tends to be larger in the model accompanied with more graph convolutional layers. That is because the over-smoothing issue gets worse with the increase in layer number. The group normalization is much more required to separate the node representations of different classes.

## 6 Instance Information Gain

In this work, we adopt kernel-density estimators (KDE), one of the common non-parametric approaches, to estimate the mutual information between input feature and representation vector [13, 14]. A key assumption in KDE is that the input feature (or output representation vector) of neural networks is distributed as a mixture of Gaussians. Since a neural network is a deterministic function of the input feature after training, the mutual information would be infinite without such assumption. In the following, we first formally define the Gaussian assumption, input probability distribution and representation probability distribution, and then present how to obtain the instance information gain based on the mutual information metric.

**Gaussian assumption.** In the graph signal processing, it is common to assume that the collected input feature contains both true signal and noise. In other word, we have the input feature as follows: $x_v = \bar{x}_v + \epsilon_x$. $\bar{x}_v$ denotes the true value, and $\epsilon_x \sim \mathcal{N}(0, \sigma^2 \boldsymbol{I})$ denotes the added Gaussian noise with variance $\sigma^2$. Therefore, input feature $x_v$ is a Gaussian variable centered on its true value.

**Input probability distribution.** We treat the empirical distribution of input samples as true distribution. Given a dataset accompanied with $n$ samples, we have a series of input features $\{x_1, \cdots, x_n\}$ for all the samples. Each node feature is sampled with probability $1/|\mathcal{V}|$ following the empirical uniform distribution. Let $|\mathcal{V}|$ denotes the number of samples, and let $\mathcal{X}$ denote the random variable of input features. Based on the above Gaussian assumption, probability $P_{\mathcal{X}}(x_v)$ of input feature $x_v$ is obtained by the product of $1/|\mathcal{V}|$ with Gaussian probability centered on true value $\bar{x}_v$.

**Representation probability distribution.** Let $\mathcal{H}$ denote the random variable of node representations. To obtain probability $P_{\mathcal{H}}(h_v)$ of continuous vector $h_v$, a general approach is to bin and transform $\mathcal{H}$ into a new discrete variable. However, with the increasing dimensions of $h_v$, it is non-trivial to statistically count the frequencies of all possible discrete values. Considering the task of node classification, the index of largest element along vector $h_v \in \mathbb{R}^{C \times 1}$ is regarded as the label of a node. We propose a new binning approach that labels the whole vector $h_v$ with the largest index $z_v$. In this way, we only have $C$ classes of discrete values to facilitate the frequency counting. To be specific, let $\mathbf{P_c}$ denote the number of representation vectors whose indexes $z_v = c$. The probability of a discrete variable with class $c$ is given by: $p_c = P_{\mathcal{H}}(z_v = c) = \frac{\mathbf{P_c}}{\sum_{l=1}^{C} \mathbf{P_l}}$.

**Mutual information calculation.** Based on KDE approach, a lower bound of mutual information between input feature and representation vector can be calculated as:

$$
\begin{aligned}
G_{\text{Ins}} = I(\mathcal{X}; \mathcal{H}) \quad &= \sum_{x_v \in \mathcal{X}, h_v \in \mathcal{H}} P_{\mathcal{X}\mathcal{H}}(x_v, h_v) \log \frac{P_{\mathcal{X}\mathcal{H}}(x_v, h_v)}{P_{\mathcal{X}}(x_v) P_{\mathcal{H}}(h_v)} \\[4pt]
&= H(\mathcal{X}) - H(\mathcal{X}|\mathcal{H}) \\[8pt]
&\geq -\frac{1}{|\mathcal{V}|} \sum_i \log \frac{1}{|\mathcal{V}|} \sum_j \exp\left(-\frac{1}{2}\frac{||x_i - x_j||_2^2}{4\sigma^2}\right) \\
&\quad - \sum_{c=1}^{C} p_c \left[ -\frac{1}{\mathbf{P_c}} \sum_{i, z_i = c} \log \frac{1}{\mathbf{P_c}} \sum_{j, z_j = c} \exp\left(-\frac{1}{2}\frac{||x_i - x_j||_2^2}{4\sigma^2}\right) \right].
\end{aligned}
$$

The sum over $i, z_i = c$ represents a summation over all the input features whose representation vectors are labeled with $z_i = c$. $P_{\mathcal{X}\mathcal{H}}(x_v, h_v)$ denotes the joint probability of $x_v$ and $h_v$. The effectiveness of $G_{\text{Ins}}$ in measuring mutual information between input feature and node representation has been demonstrated in the experimental results. As illustrated in Figures 1-4, $G_{\text{Ins}}$ decreases with the increasing number of graph convolutional layers. This practical observation is in line with the human expert knowledge about neighbor aggregation strategy in GNNs. The neighbor aggregation function as shown in Table 2 is in fact a low-passing smoothing operation, which mixes the input feature of a node with those of its neighbors gradually. At the extreme cases where $K = 30$ or $120$, we find that $G_{\text{Ins}}$ approaches to zero in GNN models without normalization. The loss of informative input feature leads to the dropping of node classification accuracy. However, our DGN keeps the input information during graph convolutions and normalization to some extent, resulting in the largest $G_{\text{Ins}}$ compared with the other normalization approaches.

# 7 Performance Comparison on Attributed Graphs

In this section, we report the model performances in terms of test accuracy, instance information gain and group distance ratio achieved on all the concerned datasets in Figures 1-4. We make the following observations:

- Comparing with other normalization techniques, our DGN generally slows down the dropping of test accuracy with the increase in layer number. Even for GNN models associated with a small number of layers (i.e., $G \leq 5$), DGN achieves the competitive performance compared with none normalization. The adoption of DGN module does not damage the model performance, and prevents model from suffering over-smoothing issue when GNN goes deeper.
- DGN achieves the larger or comparable instance information gains in all cases, especially for GAT models. That is because DGN keeps embedding matrix $H^{(k)}$ and prevents over-normalization

within each group. The preservation of $H^{(k)}$ saves input features to some extent after each layer of graph convolutions and normalization. In an attributed graph, the improved preservation of informative input features in the final representations will significantly facilitate the downstream node classification. Furthermore, such preservation is especially crucial for GNN models with a few layers, since the over-smoothing issue has not appeared.

- DGN normalizes each group of node representations independently to generally improve the group distance ratio, especially for models GCN and GAT. A larger value of group distance ratio means that the node representation distributions from all groups are disentangled to address the over-smoothing issue. Although the ratios of DGN are smaller than those of pair normalization in some cases upon SGC framework, we still achieve the largest test accuracy. That may be because the intra-group distance in DGN is much smaller than that of pair normalization. A small value of intra-group distance would facilitate the node classification within the same group. We will further compare the intra-group distance in scenarios with missing features in the following experiments.

Figure 1: The test accuracy, instance information gain, and group distance ratio in attributed Cora. We compare differentiable group normalization with none, batch and pair normalizations.

Figure 2: The test accuracy, instance information gain, and group distance ratio in attributed Citeseer. We compare differentiable group normalization with none, batch and pair normalizations.

Figure 3: The test accuracy, instance information gain, and group distance ratio in attributed Pubmed. We compare differentiable group normalization with none, batch and pair normalizations.

Figure 4: The test accuracy, instance information gain, and group distance ratio in attributed Coau-thorCS. We compare differentiable group normalization with none, batch and pair normalizations.

# 8 Performance Comparison in Scenarios with Missing Features

In this section, we report the model performances in terms of test accuracy, group distance ratio and intra-group distance achieved in scenarios with missing features in Figures 5-8. The intra-group distance is calculated by node pair distance averaged within the same group. Its mathematical expression is given by the denominator of Equation (3) in the main manuscript. We make the following observations:

- DGN achieves the largest test accuracy by exploring the deeper neural architecture with a larger number of graph convolutional layers. In the scenarios with missing features, GNN model relies highly on the neighborhood structure to classify nodes. DGN enables the deeper GNN model to exploit neighborhood structure with multiple hops away, and at the same time relieves the over-smoothing issue.

- Comparing with other normalization techniques, DGN generally improves the group distance ratio to relieve over-smoothing issue. Although in some cases the ratios are smaller than those of pair normalization upon SGC framework, we still achieve the comparable or even better test accuracy. That is because DGN has a smaller intra-group distance to facilitate node classification within the same group, which is analyzed in the followings.

- DGN obtains an appropriate intra-group distance to optimize the node classification task. While the over-smoothing issue results in an extremely-small distance in the model without normalization, a larger one in pair normalization leads to the inaccurate node classification within each group. That is because the pair normalization is designed to maintain the distance between each pair of nodes, no matter whether they locate in the same class group or not. The divergence of node representations in a group prevents a downstream classifier to assign them the same class label.

Figure 5: The test accuracy, group distance ratio and intra-group distance in Cora with missing features. We compare differentiable group normalization with none, batch and pair normalizations.

Figure 6: The test accuracy, group distance ratio and intra-group distance in Citeseer with missing features. We compare differentiable group normalization with none, batch and pair normalizations.

Figure 7: The test accuracy, group distance ratio and intra-group distance in Pubmed with missing features. We compare differentiable group normalization with none, batch and pair normalizations.

Figure 8: The test accuracy, group distance ratio and intra-group distance in CoauthorCS with missing features. We compare differentiable group normalization with none, batch and pair normalizations.