[Reviews · NeurIPS 2020]

Review 1

Summary and Contributions: This paper aims at tackling the over-smoothing problems of graph neural networks (GNNs) and trying to enable training deep GNNs. The authors proposed two over-smoothing metrics: Group Distance Ratio and Instance Information Gain, which quantify the over-smoothing from global (graph communities) and local (node individuals) views. Besides, one normalization mechanism called Differentiable Group Normalization was proposed to address the over-smoothing problem. Experiments on node classification tasks with models of varying depth were conducted.

Strengths: The problem that authors study is important for graph neural networks.

Weaknesses: (1) Empirical results seem to be weak compared to other works [1] aiming at tackling over-smoothing problem. According to table 1, Deep GNNs with DGN outperform those with other normalization mechanisms. However, the performance degradation still exists when the GNNs are made deeper. [1] proposed methods not from the normalization view but address the performance degradation, especially for very deep GNNs. (2) For the proposed over-smoothing metric, the Group Distance Ratio is a simply extension of the existing works [2] focusing on node pair distance. Though the idea is somewhat incremental, the proposed Differentiable Group Normalization relates it indeed. However, the Instance Information Gain employ mutual information between the input features and output representations as a metric, which seems to be somewhat weird. According to the Appendix F, the output representation is taken from the final prediction layer, which is the result of a linear transformation applied to the top hidden features. Thus, the quality of final prediction layer would influence the calculation of the metric. In other words, there seems to be various factors including the over-smoothing behavior of deep GNNs that directly influence this metric. (3) For the proposed Differentiable Group Normalization, the mechanism firstly learns a differentiable clustering for potential groups and then normalize the features within the same groups softly. However, the training process of the GNNs with DGN of different depth is not showed. Intuitively, the cluster assignment matrix is dynamically changed along the training of GNNs. It is better to show the influence of the DGN brought to the training of GNNs as a supplementary for experiments. [1] Chen, Ming, Zhewei Wei, Zengfeng Huang, Bolin Ding, and Yaliang Li. “Simple and Deep Graph Convolutional Networks.” In Proceedings of International Conference on Machine Learning 2020, 3730–3740, 2020. [2] Zhao L , Akoglu L . PairNorm: Tackling Oversmoothing in GNNs[J]. 2019.

Correctness: The claims, methods seem to be correct. However, for the empirical methodology, the necessity to evaluate the GNNs with varying depth on datasets consisting missing features is unaccounted. It is better to discuss what could be brought to researchers or practitioners to evaluate methods aiming at tackling the over-smoothing problem in the missing feature condition.

Clarity: This paper is easy to follow. But the left panel of Figure 3 is hard to read and could be improved by converting to a table.

Relation to Prior Work: This paper is missing a related work section, while simply compare the proposed differentiable group normalization with several other methods for over-smoothing problem in GNNs in Section 3.1. It is better to firstly review the over-smoothing problem in GNNs and then introduce existing understanding and methods for this problem.

Reproducibility: Yes

Additional Feedback:


Review 2

Summary and Contributions: Over-smoothing is one of the limiting factor for scaling deep GNN for discrimination learning. Motivated by this issue in GNN, this work propose interleaving GNN layers with differentiable group normalisation (DGN) layer. DGN normalises node features within the same group while distinguishing feature distributions among other groups. This is shown to slow down performance dropping with increasing stack. Further, in order to study over-smoothing issue it introduces two metrics. Experiments on standard datasets shows that DGN improves the performance of GNN on classification task.

Strengths: 1. This is the first work to introduce precise metrics for measuring over-smoothing in GNN - Group Distance Ratio and Instance information gain. Until recently, it had been empirically identified by studying the distance between node pairs. 2. Unlike the previous work, which tries tackling over-smoothing issue by using specific regularizing loss, this work proposes new differentiable parametric layer. This layer indirectly reduces over-smoothing. 3. Relative to prior work, the deeper GNN using DGN layers shows improved performance.

Weaknesses: 1. For a fix column Table 1, it is not clear whether the total improvement is only due to normalisation operation in DGN or due to increased params w.r.t. PN & BN. 2. Please discuss runtime impact. 3. Although, I see significant performance improvement for deeper stacks compared to prior work, the overall performance for deep stack still lags behind the shallow GNNs + DGN. 4. Currently, the novel contribution is defining two precise overs-smoothing metrics. The DGN layer definition is very obvious, although unconnected to these metrics. And differentiable clustering process is simple extension from differentiable graph pooling (DiffPool) module [A]. [A] Ying, Zhitao, et al. "Hierarchical graph representation learning with differentiable pooling." Advances in neural information processing systems. 2018.

Correctness: Yes.

Clarity: The paper is very well written and easy to follow.

Relation to Prior Work: Yes.

Reproducibility: Yes

Additional Feedback: 1. For S update, eq (7), have you tried using Gumbel-Softmax or Softmax with temperature param ? ############ Post Rebuttal: I agree with other reviewer that the current work should make a comparison to other recent work. Moreover, I am not convinced with the applicability of this work which is more or less useful for missing feature case. At the same time, I find merit in this work and believe other recent works should be considered contemporary to this work. Hence, I keep my score unchanged.


Review 3

Summary and Contributions: This paper proposes the differentiable group normalization (DGN) to mitigate the over-smooth issue inherent in Graph Neural Network. In order to guide the methods tackling the over-smooth issue, two metrics (i.e.Group Distance Ratio and Instance information gain) are proposed to evaluate how severe the node representations are smoothed. Besides, the paper furthermore shows the proposed DGN effectively mitigates the over-smooth issue and significantly improves the performance.

Strengths: The paper is well written. In particular, it provides very good motivations for group-wise normalization. In this way, it is really easy to follow the core idea of the paper. The proposed two metrics are important for guiding the future development of methods for tackling the over-smooth issue. The paper validates the performance of DGN with multiple datasets and multiple GNNs. The experiments are very solid.

Weaknesses: The paper has several weaknesses which I will detail below: # Contributions: (1), The main technical contribution of this paper is the differentiable group normalization (DGN). However, it seems that the exact same normalization has been already proposed -- It was called attentive normalization [1] instead. The idea of the attentive normalization is to split features into multiple groups and normalize with’s its own statistics (i.e., mean and std). From my point of view, the attentive normalization and the DGNs basically share the idea and even formulation. The minor difference would be that attentive normalization in [1] is trained with additional regularization tricks to avoid mode collapse. [2], Aslo, as mentioned in [1], it’s possible that some groups get the zero assignments in the assignment matrix -- e.g., No node is assigned in some groups. I think it would be straightforward to investigate this issue and perhaps improve performance with the same regularization tricks as in [1]. (3), There are also some other closely related works such as attentive context normalization [2] and mode normalization [3]. Both normalizations learn an assignment matrix as the proposed DGN -- split the inputs into multiple or single groups, and then normalize accordingly. I think the paper should discuss or even compare DGN with them. Reference: [1]Wang et al. “Attentive Normalization for Conditional Image Generation”. CVPR 2020. [2], Sun et al. “ACNe: Attentive Context Normalization for Robust Permutation-Equivariant Learning” CVPR 2020 [3], Deecke et al. “Mode Normalization” ICLR 2020.

Correctness: The claim of the proposed differential group normalization (DGN) might not be true as explained in the weaknesses.

Clarity: Yes, the writing of this paper is excellent.

Relation to Prior Work: The paper clearly relate with prior works on over-smooth issue of GNN. But it fails to relate with works on normalization as mentioned in the weaknesses.

Reproducibility: Yes

Additional Feedback: AFTER REBUTTAL: I have gone through other reviewers' comments and the author's feedback. Other reviewers' concerns are valid to me. And I don't think the rebuttal has solved my main concern about the overlap between attentive normalization (AN) and the proposed DGN. While the authors put the performance of AN in the table in rebuttal, they didn't clarify the difference between AN and DGN. From my point of view, they basically share the same form which was also mentioned in my review. While the performance of AN and DGN are different in as shown in the table of rebuttal, I doubt if the rebuttal has properly implemented the AN given that there are several regularization tricks in AN's paper. Therefore, I would keep my original ratings.


Review 4

Summary and Contributions: The paper targets at the over-smoothing issue in GNNs by considering the community structures in a graph in terms of the two proposed over-smoothing metrics and a differentiable group normalization. Experimental results on several data sets have validated the effectiveness of the propsoed method.

Strengths: The soundness of this research work is good from both the theoretical thinking and empirical evaluation perspectives. The claim is quite valuable for research of GNNs and very relevant to the NeurIPS community.

Weaknesses: It would be better if the paper can give a few case studies(typical examples) with explanations and analyses. For instance, for SGC on the data set Citeseer in Table 1, #K is 30. It is very big compared to other cases (in some cases #K is 2). It's not easy to imagine this. What happens? and how? Futher, why for all three models on Coauthors, #K is always 1 however, in Table 2, #K turns quite big, ranging from 15 ot 25? This might be a confusing for readers.

Correctness: The claims and method are correct and, the empirical methodology is correct.

Clarity: Yes, the paper is well written.

Relation to Prior Work: Yes, the paper clearly discussed how this work differs from previous contributions.

Reproducibility: Yes

Additional Feedback: Concrete analysis on experimental results is expected.

[Author Response · NeurIPS 2020]

We thank all the reviewers for the constructive comments. We've done additional experiments to address the concerns
about the comparison with very recent efforts published this year, and clarified the confusions one-by-one. We reiterate
our contributions: (1) Global & local metrics; (2) general DGN to improve GNNs even comparing with recent efforts.

**To Reviewer 1.** Q: Comparison with GCNII$^{*1}$. We explore to relieve the over-smoothing issue from the perspective
of group normalization, and propose a general module – DGN. It can improve different GNN backbones including
GCNII$^*$. Our experiments focus on the comparison among existing normalization approaches, rather than fine tuning
hyperparameters for state-of-the-art performance. The mentioned paper[1] appeared (July 2020) after NeurIPS submission.
We will include the comparison results on backbone network GCNII$^*$ in the revised version. Table 1 presents part of
the results, where DGN consistently outperforms the other normalization approaches for different network depths. NN
(none normalization) denotes the original GCNII$^*$ model. It could be further improved by DGN.

Table 1: The test accuracy in percentage over backbone network GCNII$^*$ on the dataset Cora.

| # Layers | NN | BN | PN | DGN | # Layers | NN | BN | PN | DGN | # Layers | NN | BN | PN | DGN |
|---|---|---|---|---|---|---|---|---|---|---|---|---|---|---|
| 2 | 80.2 | 76.5 | 74.2 | **83.0** | 16 | 83.5 | 74.4 | 73.3 | **85.1** | 64 | 85.3 | 73.7 | 68.1 | **86.4** |

Q: Metrics. (1) Group distance ratio measures the over-smoothing based on the community impact on the preferred node
distance, which has not been studied before. This intuitive metric provides us an empirical guidance to design DGN
module to improve model performance. (2) As mentioned in Section 2.3, both over-fitting and over-smoothing influence
metric $G_{\mathrm{Ins}}$ of instance information gain. For SGC model, we project node features into label space in the first layer
and remove all the following trainable matrices. The output representation is generated from the final convolutional
layer to eliminate the impact of prediction layer. We observe that $G_{\mathrm{Ins}}$ still decreases with more layers, which validates
the rationality of our metric tailored to measure the mutual information between graph signals. Indeed, GCN/GAT
usually transforms features into label space in the final convolutional layer, and removes the redundant prediction layer.

Q: Training process. Thanks for pointing it out. We find that DGN helps GNN models converge to a better train-
ing/validation loss quickly within hundreds of epochs. Due to space limit, we will include it in the revised version.

Q: Missing features. The current benchmarks associated informative node attributes usually require shallow GNNs. But
the over-smoothing issue only appears in the scenarios necessary to stack deeper GNNs to access useful neighbors many
hops away. As discussed in Section 3.3, one of such scenarios, missing features, exists in many real-world applications
and has been studied before. Our DGN is shown to effectively tackle this issue and enable deeper neural architectures.

**To Reviewer 2.** Q: Increased params. Thanks for the comment. In GNNs, the increased parameters may instead lead to
over-fitting. As illustrated in Figure 3 and indicated by group distance ratio, the performance improvement of DGN is
indeed brought by soft clustering to disentangle unrelated nodes, comparing with the single group in PN & BN.

Q: Deeper vs shallow models. The deeper model unleashes power in the scenario where a node really needs information
many hops away to better represent its label. We study such a complex scenario with missing node features in Section
3.3. As also shown in Table 2 and Appendix, deeper GNNs+DGN achieve better performance than the shallow ones.

Q: Softmax. We've tried Softmax temperatures and regularizations to enforce that each node is assigned to one group.
We empirically found that soft clustering itself can achieve the similar objective and reduce computation complexity.

**To Reviewer 3.** Q: Comparing attentive normalization. Thanks for pointing it out, and we will include the comparison
in the revised version. Here we present part of results of attentive normalization[2] (AN), attentive context normalization[3]
(ACN) and our DGN in Table 2, where DGN consistently outperforms them in all cases. This is because ACN has
only one group and AN additionally samples random groups during model inference. They are not in line with the
transductive node classification task where the underlying graph has a series of fixed community structures. Furthermore,
the motivations for our DGN and AN/ACN are different. While AN/ACN target at capturing the long-range relations
between pixels, DGN intends to improve the distance between different groups to mitigate the over-smoothing issue.

Table 2: Test accuracy in percentage over GCN model for dataset Cora.

| # Layers | AN | ACN | DGN | # Layers | AN | ACN | DGN | # Layers | AN | ACN | DGN |
|---|---|---|---|---|---|---|---|---|---|---|---|
| 2 | 77.1 | 74.4 | **82.0** | 15 | 16.3 | 42.8 | **75.2** | 30 | 17.2 | 20.9 | **73.2** |

Q: Zero assignment. A similar case happens in the right part of Figure 3 in our paper, where only 6 out of 10 groups are
assigned with nodes. In the hyperparameter study, we further find that this issue could be solved by simply choosing a
proper group number close to the class categories. We will test the regularization term in paper[2] in the revised version.

**To Reviewer 5.** Q: Case studies. We appreciate that you found the work valuable and pointed out this question. As
mentioned in Section 2.3, SGC achieves better performance with a larger value of $K$, since the simplifying mechanism
avoids over-fitting to enable layer stacking. Compared with Table 1, Table 2 presents results on the complex scenarios
with missing features. As explained in Section 3.3, the deeper models would be preferred since we need to exploit the
distant neighbors to collect sparse information. We will explain them in detail in the revised experiment section.

[1]Chen et al. "Simple and Deep Graph Convolutional Networks," ICML 2020. [Available in arXiv on 4 Jul 2020]
[2]Wang et al. "Attentive Normalization for Conditional Image Generation," CVPR 2020. [Available in arXiv on 8 Apr 2020]
[3]Sun et al. "ACNe: Attentive Context Normalization for Robust Permutation-Equivariant Learning," CVPR 2020. [23 Apr 2020]


[Meta-Review · NeurIPS 2020]

This paper presents a method to address the over-smoothing issue in deep graph neural network with differentiable group normalization and metrics to measure over-smoothing. All reviewers agree that this paper tackles an important problem and the empirical results verify the main claim of the paper. The reviewers raised some issues regarding comparisons with previous work. However, the authors noted that these papers are relatively recent (available publicly in April 2020 or later). I consider these papers to be parallel work and therefore understand why the authors did not compare with them in the current version. I encourage the authors to include these comparisons for completeness in the final version, as well as other feedbacks around analysis from R2 and R5.